# The Efficacy of Additives for the Mitigation of Aflatoxins in Animal Feed: A Systematic Review and Network Meta-Analysis

**DOI:** 10.3390/toxins14100707

**Published:** 2022-10-15

**Authors:** Oluwatobi Kolawole, Wipada Siri-Anusornsak, Awanwee Petchkongkaw, Julie Meneely, Christopher Elliott

**Affiliations:** 1Institute for Global Food Security, National Measurement Laboratory: Centre of Excellence in Agriculture and Food Integrity, Queen’s University Belfast, 19 Chlorine Gardens, Belfast BT9 5DL, UK; 2The International Joint Research Center on Food Security (IJC-FOODSEC), 113 Thailand Science Park, Pahonyothin Road, Khong Luang, Pathum Thani 12120, Thailand; 3Department of Food Science and Technology, Faculty of Agro-Industry, Kasetsart University, Bangkok 10900, Thailand; 4Scientific Equipment and Research Division, Kasetsart University Research and Development Institute (KURDI), Kasetsart University, Bangkok 10900, Thailand; 5School of Food Science and Technology, Faculty of Science and Technology, Thammasat University, 99 Mhu 18, Pahonyothin Road, Khong Luang, Pathum Thani 12120, Thailand

**Keywords:** aflatoxins, binders, network meta-analysis, additives, mitigation, livestock, clay minerals, antioxidant, yeast cell wall

## Abstract

The contamination of animal feed with aflatoxins is an ongoing and growing serious issue, particularly for livestock farmers in tropical and subtropical regions. Exposure of animals to an aflatoxin-contaminated diet impairs feed efficiency and increases susceptibility to diseases, resulting in mortality, feed waste, and increased production costs. They can also be excreted in milk and thus pose a significant human health risk. This systematic review and network meta-analysis aim to compare and identify the most effective intervention to alleviate the negative impact of aflatoxins on the important livestock sector, poultry production. Eligible studies on the efficacy of feed additives to mitigate the toxic effect of aflatoxins in poultry were retrieved from different databases. Additives were classified into three categories based on their mode of action and composition: organic binder, inorganic binder, and antioxidant. Moreover, alanine transaminase (ALT), a liver enzyme, was the primary indicator. Supplementing aflatoxin-contaminated feeds with different categories of additives significantly reduces serum ALT levels (*p* < 0.001) compared with birds fed only a contaminated diet. Inorganic binder (P-score 0.8615) was ranked to be the most efficient in terms of counteracting the toxic effect of aflatoxins, followed by antioxidant (P-score 0.6159) and organic binder (P-score 0.5018). These findings will have significant importance for farmers, veterinarians, and animal nutrition companies when deciding which type of additives to use for mitigating exposure to aflatoxins, thus improving food security and the livelihoods of smallholder farmers in developing countries.

## 1. Introduction

Aflatoxins are a group of naturally occurring mycotoxins produced by fungal species belonging to *Aspergillus* section *Flavi*, particularly *A. flavus* [1,2]. More than 20 aflatoxin molecules have been identified to date; the most important ones from a food and feed safety perspective are the difurocoumarocyclopentenone group (aflatoxin B1 and aflatoxin B2) and the difurocoumarolactone group (aflatoxin G1 and aflatoxin G2) [1,2]. Aflatoxin contamination of feed and feed ingredients is a global problem, especially in tropical and subtropical regions where warm temperatures and humidity favour the growth of aflatoxin-producing fungi [1,2]. Furthermore, agricultural practices such as poor harvesting and improper crop storage significantly influence aflatoxin contamination of feed raw materials such as wheat, soybean, and maize [2,3].

Livestock’s dietary exposure to feed contaminated with aflatoxins result in a wide range of adverse health effects, including a significant alteration in biochemical, haematological, and performance parameters [4,5]. Monogastric animals, including pigs and poultry, are the farm animals most sensitive to the toxic effects of aflatoxins [6,7]. Ruminants are more resistant than non-ruminant animals because the rumen microbiota can degrade or deactivate toxins [6,7]. Of all the aflatoxin molecules, AFB1 is the most prevalent in feedstuffs and recognised as the most hazardous due to its genotoxic and carcinogenic potential [4]. The liver is the primary target organ for aflatoxins. Metabolism of these compounds in the liver by cytochrome P450 enzymes results in the formation of aflatoxin B- 8-9-epoxide (AFBO)—a major carcinogenic metabolite of aflatoxins [8]. AFBO is very unstable and spontaneously reacts to form adducts with DNA, RNA, and proteins, leading to hepatic cell and tissue injury as well as pro-inflammation and oxidative stress, which promotes liver damage, reduced feed intake, body weight, and mortality [8]. Besides the adverse health effects of aflatoxins on animal health, welfare, and productivity, aflatoxin-contaminated feed also poses a food safety concern for humans as aflatoxin M1 (a possible human carcinogen) is excreted in milk [9]. Moreover, long-term exposure of farm animals to contaminated feeds can lead to the carry-over of aflatoxins to animal-derived foods, including meat and egg [10].

In regions, particularly Sub-Saharan Africa and Southeast Asia, where poverty and food insecurity are increasing due to a wide range of factors, aflatoxins have to date been neglected as being a significant contributing factor. Aflatoxin occurrences of between 25% and 100% have been reported in livestock feed and feed ingredients in these regions at levels above the European Union (EU) regulatory limit of 20 μg/kg [11,12,13]. The annual cost of aflatoxin contamination in the Philippines, Thailand, and Indonesia was estimated to be nearly USD 1 billion [14]. This study is nearly 30 years old, and the costs are likely substantially higher. Similarly, African countries incur losses of several millions of dollars annually due to aflatoxin contamination of agricultural products [15]. These substantial economic losses are largely associated with costs incurred from surveillance, handling and testing of feed lots/batches, feed waste due to mortality and reduced animal performance, product rejection mostly due to non-compliance with the established regulatory limits, and veterinary care to improve health and productivity of farm animals. Therefore, the health consequences and economic importance of dealing with aflatoxins in feed need to be better understood in terms of mitigating one of the impacts of climate change on food security.

A wide range of feed additives with different compositions and modes of action have been developed and evaluated over the last decades as a post-harvest dietary intervention strategy to curtail exposure to aflatoxins [16,17,18]. These additives can be divided into three groups: binders, modifiers, and antioxidants [16]. Mycotoxin binders prevent the absorption of mycotoxins from the gastrointestinal tract of livestock animals by adsorbing the toxins to their surface to form a binder-mycotoxin complex, which is later excreted in the faeces. Examples of binders include clay minerals, yeast-cell walls, polymers, and agricultural waste products [16]. Mycotoxin modifiers are of biological origin (bacteria, fungi, and enzymes); they can modify (bio-transform) the chemical structure of mycotoxins to yield metabolites that are less toxic to non-toxic compared to the parent compounds. Antioxidants do not bind nor modify mycotoxins, but they can counteract the toxic effects of aflatoxins by targeting oxidative stress and inflammatory signalling pathways to prevent aflatoxin-induced toxicities [16].

Due to the abundance and different composition and modes of action, as well as claims of performance of these agents, it is challenging for farmers, animal nutrition companies, and feed processors to select and utilise the most suitable agents [17]. In addition, because of socioeconomic status and lack of knowledge of farmers in low-income countries regarding the detrimental impact of aflatoxins and available intervention strategies, animals are often fed contaminated feed [13]. This contributes to poor animal health, low profit margins, and feed waste. Therefore, this study conducted a systematic review of aflatoxin intervention studies in livestock to compare and identify the most efficient agent for mitigating aflatoxin toxicity in livestock using a frequentist network meta-analysis. Following the initial database search, the majority of studies retrieved were focused on poultry species, with few studies available for pigs and ruminants. Therefore, to prevent publication bias, we conducted only the network meta-analysis of poultry studies. Network meta-analysis allows the comparisons of the effects of multiple treatments on a health outcome. Moreover, it allows for a quantitative synthesis by combining direct and indirect evidence from comparisons of treatments within experimental trials based on a common comparator, which in this study is a control feed or aflatoxin-free diet.

## 2. Results

### 2.1. Search Results

The database search identified 1010 articles from the Web of Science, PubMed, and Scopus. These studies were assessed for inclusion and exclusion using the pre-specified eligibility criteria. The title and abstract of 323 articles were assessed following the removal of 687 duplicates. Of the 145 articles examined for full texts, only 31 met the inclusion criteria and were selected for network meta-analysis. The number of included and excluded articles at different phases of the selection process are presented as a flowchart in Figure 1.

### 2.2. Characteristics of the Included Studies

Most of the included articles were field and randomised controlled studies conducted in Asia (68.5%), followed by Africa (20%), America (8.5%), and Europe (3%). In total, 2752 birds (mostly broiler chickens) were used for the trials. They were randomly distributed into three feeding groups, namely negative control (aflatoxin-free feed), positive control (aflatoxin-contaminated feed), and intervention (aflatoxin-contaminated feed + additive). While the levels of aflatoxins in groups exposed to contaminated feed and contaminated feed plus additives ranged from 50 µg/kg to 2500 µg/kg, no aflatoxins were detected in feeds given to animals in the control group (<0 µg/kg). The mean aflatoxin levels in contaminated feed supplemented with antioxidants, inorganic binders, and organic binders were 744 µg/kg, 855 µg/kg, and 839 µg/kg, respectively. Animals were fed *ad libtum*, and the average duration of trials was 25 days. The background characteristics of the included studies are summarized in Table 1.

### 2.3. Study Classification

The selected studies reported different feed additives ranging from clay minerals, such as bentonite, zeolite, kaolin, and humic acid, to polymers, yeast, and bacterial cell walls, as well as plant extracts, vitamins, and essential oils (Table 1). These agents were grouped into three categories based on their composition and mode of action: inorganic binder, organic binder, and antioxidant. Studies with additives that could not be classified under these groups were removed from the analysis. Out of the 31 studies found eligible, eleven studies investigated the efficacy of three or more different feed additives (Table 1). Thus, 43 trials or paired groups (control vs. intervention) were included in the network meta-analysis. Out of the 43 trials, 18, 13, and 12 trials were grouped, respectively, under antioxidants, inorganic binder, and organic binder (Table 1). The percentage number of trials for each additive category is shown in Figure 2. The type of additive and inclusion level used in each trial are presented in Table 1.

### 2.4. Meta-Analysis Results

#### Effect of Aflatoxins on Serum Alanine Transaminase

Thirty-one studies comprised of 43 trials evaluated the effect of aflatoxin-contaminated feed on poultry serum ALT. Conventional pair-wise meta-analysis of pooled mean difference (MD) was 8.83 with 95% Confidence Intervals (CIs) of 6.9 to 10.7; *p* < 0.0001, suggesting a significant increase in serum ALT of birds fed aflatoxin-contaminated feed compared with birds given negative or aflatoxin-free diet (Figure 3). No significant heterogeneity was observed (I^2^ value = 26%; *p* = 0.06), thus, subgroup analysis was not performed. Moreover, the symmetrical distribution of the funnel plot and Egger’s test *p*-value (*p* = 0.68) suggested no risk of publication bias (Figure 4).

### 2.5. Efficacy of Feed Additives to Mitigate Aflatoxins

In total, 26 different feed additives were used and were classified into three groups based on modes of action and composition (Table 1). The efficacy of different additives estimated using MD with 95% CIs is displayed in Figure 5. As indicated in the results, compared to control groups, all the feed additive groups effectively mitigated the toxic effect of aflatoxins. Supplementation of feed with antioxidant (MD = −1.82 [95% CI = −3.45 to −0.18; *p* = 0.04); inorganic binder −2.83 (95% CI: −5.28 to −0.35; *p* = 0.02), and organic binder −1.53 (95% CI: −3.19 to 0.14; *p* = 0.05), led to a significant reduction in serum levels of ALT in birds fed contaminated diet. The efficacy of the additives was ranked using P-scores derived from the network point estimates. The P-score was highest for the inorganic binder (P-score 0.8615), followed by the antioxidant (P-score 0.6159) and organic binder (P-score 0.5018). There was a statistically significant difference between the efficacy of inorganic binders and antioxidants (*p* < 0.05) as well as organic binders (*p* < 0.05). However, no significant difference was observed between the efficacy of antioxidants and organic binders (*p* > 0.05). The percentage of direct and indirect evidence used for each estimated comparison and the random effect network estimates for all treatment comparisons, with effect sizes and CIs, are presented in Appendix A. Moreover, the network structure graph of comparisons of different categories of feed additives is shown in Appendix A.

The tests of heterogeneity (within experimental designs) and total inconsistency (between experimental designs) were used to assess the validity or consistency of the frequentist model based on the full design-by-treatment interaction random effects model. There was no evidence suggesting inconsistencies between direct and indirect comparisons in the results of our network meta-analysis (Appendix A). Neither the within-design heterogeneity nor between-design inconsistency were significant (ps > 0.977). Moreover, the Q statistics for assessing homogeneity was not statistically significant (Q = 0.2; *p* = 0.979; tau^2^within = 0) (Appendix A), indicating consistency for head-to-head comparisons. In addition, the symmetry of the comparison-adjusted funnel plot for the network meta-analysis and Egger’s test *p*-value (0.30) indicated no publication bias (Figure 6).

## 3. Discussion

The recent global mycotoxins surveys showed that over 23% of finished feed and feed ingredients were contaminated with aflatoxins [11]. Aflatoxins feed crop contamination varies significantly from region to region, with countries in the tropics and subtropics (mostly Southeast Asia, Africa, and South and Central America) having higher concentrations compared to the rest of the world (Figure 7) [11,13]. Reliance on smallholdings or family-operated farms in most of these regions not only plays an important role in rural communities and their agriculture but also plays a crucial role in food security. According to the Food and Agriculture Organization, small family farmers produce a third of the world’s food [44]. Besides the health consequence of aflatoxin-contaminated feed ingredients, this problem also raises concerns for food safety and food security. Nutritionally, chicken provides a rich source of high-quality protein. It is also relatively low in fat and contains significant levels of monounsaturated fatty acids and polyunsaturated fatty acids, as well as numerous vitamins, essential minerals, and amino acids [45]. The increasing contamination of feed, exacerbated by climate change, will influence the health and productivity of animals, affecting the supply of safe and nutritious food [11,12,13]. Therefore, it is crucial that reliable and effective mitigations are in place.

Several additives have been developed and evaluated to help minimise livestock exposure to aflatoxins and improve animal health and performance [16,46]. There are very limited studies investigating the efficacy of additives to mitigate aflatoxin toxicity in ruminants and pigs; thus, this study focused solely on poultry. An extensive literature search and systematic review were conducted to identify eligible studies. Only studies that exposed poultry birds to natural aflatoxin-contaminated feed were included in our network meta-analysis. This is because naturally contaminated feed is more toxic than pure compounds due to the synergistic effects of different types of aflatoxins [16,47]. Additionally, because there are very few studies that monitored the aflatoxin biomarker, ALT, an enzyme found primarily in the liver, was selected as the primary indicator of aflatoxin exposure and the efficacy of the additives [48]. ALT exists in the blood at a very low concentration. However, when the liver is damaged (i.e., following aflatoxin exposure), ALT is released into the blood, resulting in an elevated concentration. This often precedes more apparent symptoms such as reduced body weight and feed refusal. Thus, ALT is considered an early and specific marker of hepatocellular damage [26,48].

The studies that met the pre-determined eligible criteria were meta-analysed using a frequentist network approach. Frequentists define and interpret an event’s probability in terms of how often it is expected to occur if repeated [49]. The results of our pair-wise meta-analysis revealed that compared to the control group, exposure of poultry birds to an aflatoxin-contaminated diet significantly increased serum ALT levels (up to 53%, MD = 8.83 [95% CI = 6.9 to 10.7]; *p* < 0.0001). All 43 trials included in our meta-analysis reported a significant increase in poultry ALT levels following aflatoxin exposure, with severity dependent on the duration of exposure. Regarding the efficacy of additives supplemented with the feeds to restore or normalise serum ALT levels, all the additives evaluated were found to reduce serum ALT levels compared with the control group and birds fed only an aflatoxin-contaminated diet. The additives were ranked according to their level of effectiveness using the frequentist P-score values derived from the frequentist network point estimates. The P-score measures the extent of certainty that one intervention is better than another on a scale from 0 (worst) to 1 (best) [50]. The inorganic binder was found to be the most efficient (P-score 0.8615), followed by antioxidant (P-score 0.6159) and organic binder (P-score 0.5018).

Inorganic binders are aluminosilicates composed of silica, alumina, and significant amounts of alkaline and alkaline earth ions [17]. Due to a high cation exchange capacity, they can bind or adsorb aflatoxins to their interlayer spaces, external surfaces, and edges through a different mechanism of action, including chemisorption and ion exchange, i.e., between clay cations and aflatoxin carbonyl groups [17,51]. Several clay minerals with aflatoxin-binding capacity have been patented, with some commercially available as feed additives for livestock farmers and animal nutrition companies. Clay minerals are low-cost and widely available [16,51]. However, their effectiveness varies depending on source and composition. There are also concerns about the safety and stability of these agents in the gastrointestinal tract and their effect on the palatability of animal feed. Moreover, their non-specific binding activity can negatively affect the bioavailability of essential nutrients and veterinary medicine [16,17].

Antioxidants are additives supplemented with feed primarily for prolonging the shelf life and sensory qualities, preventing rancidity, and the oxidation of critical nutrients such as pigments and vitamins. Antioxidants do not bind or modify mycotoxins, but they can counteract the toxic effects of mycotoxins by targeting oxidative stress and inflammatory signalling pathways to alleviate aflatoxin-induced toxicities [19,37]. Aflatoxins inhibit the expression of antioxidant enzymes, including superoxide dismutase and glutathione peroxidase resulting in the overproduction of free radicals or reactive oxygen species (ROS) [8,25]. When the production of ROS exceeds the antioxidant capacity of a cell, several intracellular mechanisms that promote cell death and oxidative damage to DNA, proteins, and membrane lipids (lipid peroxidation) are activated. Many natural antioxidants have been demonstrated to act as free radical scavengers, conferring beneficial or protective effects against toxic effects induced by aflatoxins [19,25,37]. Generally, most antioxidants activate the nuclear factor erythroid 2-related factor 2(Nrf2)-associated genes leading to the release of the battery of detoxification and antioxidant enzymes, which protect the cells from aflatoxin-induced inflammatory and oxidative damage [52]. The antioxidants used in studies selected for our meta-analysis are outlined in Table 1.

Organic binders are mostly yeast and lactic and propionic acid bacteria cell walls and polymers such as glucomannan. The yeast cell wall (*Saccharomyces cerevisiae*) comprises lipids, chitins, protein, and polysaccharides such as mannan and glucan [38]. The reticular organization of β-D-glucans and the distribution between β-(1,6)-D-glucan and β-(1,3)-D-glucan play a significant role in the adsorption of aflatoxins [16,38]. The binding potential of lactic and propionic acid bacteria (mostly *Propionibacterium and Lactobacillus* species) is attributed to peptidoglycan and teichoic acid, a glycopolymer, which is embedded within the peptidoglycan layers [16]. The mechanism of aflatoxin binding by the cell walls of yeast and lactic acid bacteria is mainly through ionic and hydrophobic interactions [16]. Glucomannan, a water-soluble polysaccharide, has also been shown to be a good aflatoxin adsorbent [39].

In contrast to antioxidants, inorganic, and organic binders, studies on *in vivo* efficacy of modifiers or detoxifiers (i.e., probiotics and enzymes) are scarce in the scientific literature. The few available studies [53,54] were not sufficient for meta-analysis, thus, were excluded to avoid publication bias. The application of probiotics and enzymes in mycotoxin mitigation in farm animals is limited because microbes/enzymes are mostly active in their strict environment. Moreover, the cost of production is high, and it is very difficult to evaluate the mechanism of biotransformation and the identification of metabolites and their toxicities [16]. Nevertheless, they offer an alternative environmentally friendly strategy for minimising livestock exposure to aflatoxins.

## 4. Conclusions

To our knowledge, the present study is the first meta-analysis to investigate the efficacy of different categories of feed additives to minimise poultry exposure to aflatoxins. The network meta-analysis enabled us to, directly and indirectly, compare the effectiveness of three different categories of feed additives that had previously been evaluated for their potential to mitigate poultry exposure to aflatoxins. This study revealed that supplementing aflatoxin-contaminated feed with antioxidants (such as vitamins and plant extracts), inorganic binders (including bentonite and aluminosilicates), and organic binders (lactic acid bacteria & yeast cell walls and polymers) counteracted the toxic effect of aflatoxins in poultry birds. In terms of performance, inorganic binders provided more protection than antioxidants and organic binders. The importance of these findings needs to be conveyed to stakeholders, especially in the regions most affected by aflatoxins.

## 5. Materials and Methods

### 5.1. Search Strategy and Eligibility Criteria

Preferred Reporting Items of Systematic reviews and Network Meta-Analyses (PRISMA-NMA) recommendations were followed for literature search, study selection, and data extraction. Since this review did not involve any animal trials, ethical approval was not required. The following databases were searched: PubMed, Scopus, and Web of Science, from inception to April 2022 and updated in May 2022, to identify additional publications. The search keywords for each database are provided in Appendix A. Eligibility criteria were defined based on PICOS (population, intervention, comparator, outcome, and study) design approach:

Population (P): Poultry birds fed natural aflatoxin-contaminated and aflatoxin-free feed.

Intervention (I): The eligible interventions were additives or products for counteracting the toxic effects of aflatoxins in poultry birds.

Comparator (C): Negative control, aflatoxin-free or control feed.

Outcome (O): Serum level of alanine transaminase (ALT). ALT is a liver enzyme released into the blood following liver damage. Thus, it represents a good marker for accurately predicting aflatoxins exposure.

Study designs (S): Field studies were eligible for inclusion. Meanwhile, observational studies without a comparator group (control) were excluded.

Additionally, all eligible studies must indicate the type and levels of aflatoxins in feed. The duration of the feeding trial must be at least 21 days. Conference abstracts, proceeding papers, editorials, commentaries, study protocols, and reviews were excluded. Furthermore, studies with no full text and incomplete data were not included.

### 5.2. Study Selection Process and Screening

All references were exported to Mendeley and then transferred to Covidence (https://www.covidence.org/, accessed on 1 April 2022) for deduplication, screening, and data extraction. To minimise the likelihood of expunging the potentially relevant studies, two reviewers independently screened the title and abstracts against the pre-defined eligibility criteria. Full texts of the eligible articles were then screened by the same authors to exclude the irrelevant studies. Any disagreements were resolved by consensus.

### 5.3. Data Extraction and Assessment of Risk of Bias

All the studies included were thoroughly assessed for quality to ensure compliance with the eligibility criteria for this review before being included. Data were extracted independently from trials that met the inclusion criteria of a pre-defined data extraction sheet. The extracted data included: (a) bibliographic and general information (including author, title, publication year, and location); (b) details of animals used in each study/trial (including sample size, age, and breed); (c) aflatoxins levels in control and treatment groups; (d) intervention (type of additive (single/combined), dose of supplementation and duration); (e) mean serum level of ALT, with standard deviations. When the mean and standard deviation of serum ALT were reported as graphs, the data were extracted using WebPlotDigitizer Version 4.4 (Pacifica, CA, USA). When the data was missing or unclear, relevant authors were contacted to obtain the necessary information. Two researchers independently assessed the quality of the eligible studies selected for meta-analysis to eliminate biases such as incomplete outcome data, statistical methods, and selective reporting of results. All the included trials were of low risk of bias. Extracted data were cross-verified for typographic errors and accuracy.

### 5.4. Data Analysis

The primary outcome was estimated as a mean difference (MD) with a 95% confidence interval (CI). A pair-wise meta-analysis was performed to make a direct comparison between control and aflatoxin-contaminated feed and interventions. In addition to traditional meta-analysis, a network meta-analysis was performed using the random effect model within a frequentist framework. Heterogeneity (within designs) and total inconsistency (between designs) were used to assess the consistency of our network model. Heterogeneity was assessed using the Q and I-squared statistics, and was considered low, moderate, or high for I-squared values of <25%, 25% to 50%, and >50%, respectively. The netsplit function was used to assess inconsistencies between the direct and indirect comparisons and visualised using a forest plot. The P-score values for ranking the efficiency of the additives were calculated using the netrank function, with a forest plot used to visualise the pooled treatment comparison. A comparison-adjusted funnel plot and Egger’s regression test were used to evaluate the risk of publication bias. All the data analyses were performed with “meta”, “metafor”, and “dmetar” packages in the statistical program R version 4.2.0.

## Figures and Tables

**Figure 1 toxins-14-00707-f001:**
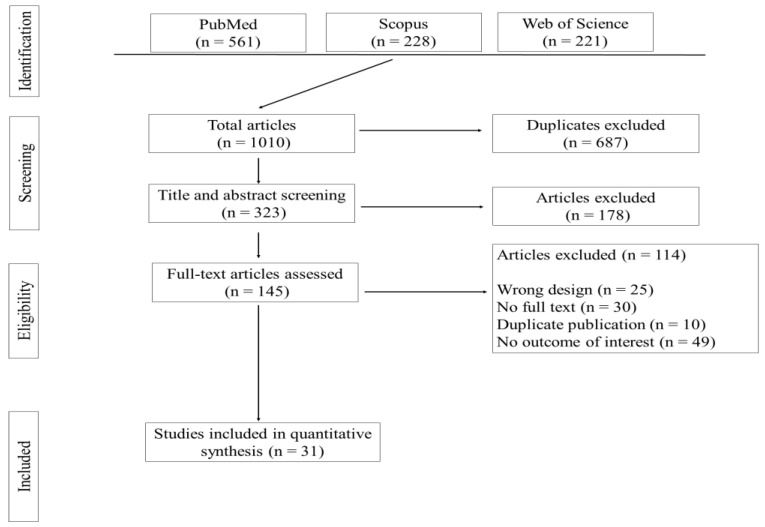
A summary of the screening and selection process to retrieve eligible studies for network meta-analysis.

**Figure 2 toxins-14-00707-f002:**
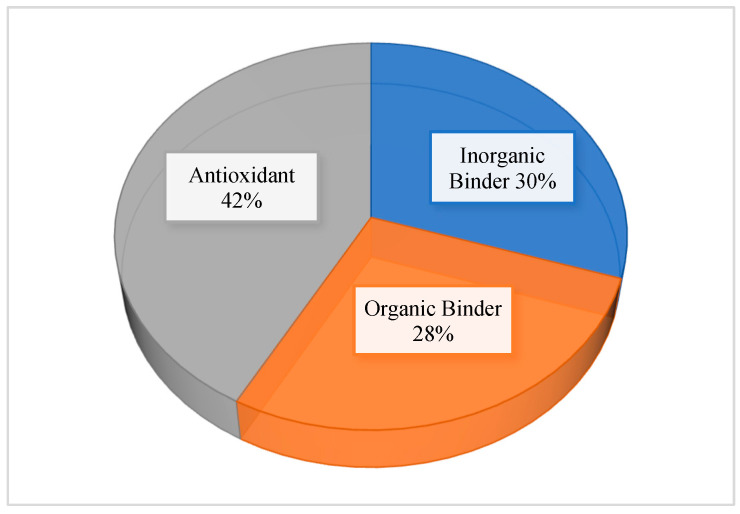
Percentage number of trials in each additive group: antioxidants (18/43); inorganic binder (13/43); organic binder (12/43).

**Figure 3 toxins-14-00707-f003:**
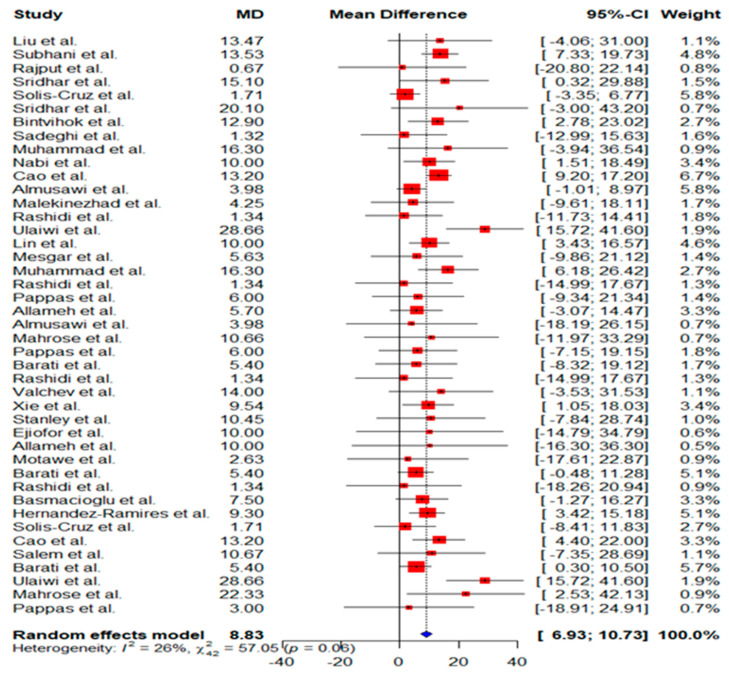
Random effect model forest plot of serum level of alanine transaminase in poultry birds fed natural diet contaminated with and without aflatoxins. Each data marker represents a study, and the size of the red box is proportional to the weight of the study. The summary effect size is denoted by the blue diamond. References: Liu et al. [29], Subhani et al. [26], Rajput et al. [28], Sridhar et al. [30], Solis-Cruz et al. [25], Sridhar et al. [32], Bintvihok et al. [34], Sadeghi et al. [31], Muhammad et al. [33], Nabi et al. [21], Cao et al. [8], Almusawi et al. [24], Malekinezhad et al. [22], Rashidi et al. [23], Ulaiwi et al. [27], Lin et al. [19], Mesgar et al. [20], Muhammad et al. [33], Rashidi et al. [23], Pappas et al. [44], Allameh et al. [5], Almusawi et al. [24], Mahrose et al. [41], Pappas et al. [44], Barati et al. [7], Rashidi et al. [23], Valchev et al. [42], Xie et al. [4], Stanley et al. [40], Ejiofor et al. [36], Allameh et al. [5], Motawe et al. [38], Barati et al. [7], Rashidi et al. [23], Basmacioglu et al. [39], Hernandez-Ramires et al. [36], Solis-Cruz et al. [25], Cao et al. [8], Salem et al. [37], Barati et al. [7], Ulaiwi et al. [27], Mahrose et al. [41], Pappas et al. [44].

**Figure 4 toxins-14-00707-f004:**
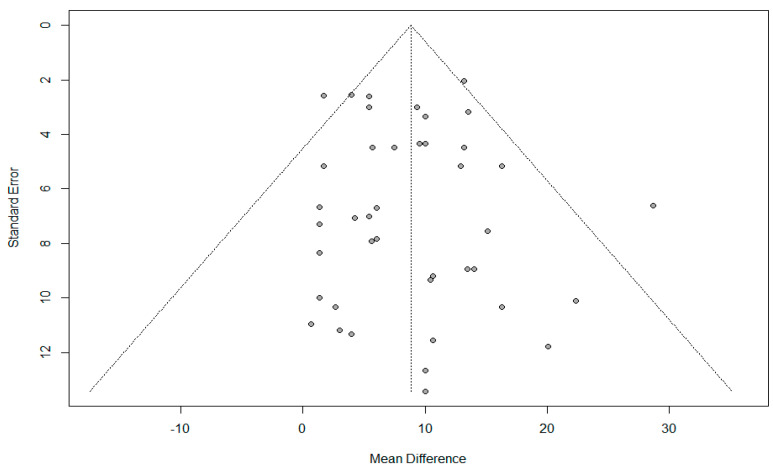
Funnel plot of serum level of alanine transaminase in poultry birds fed natural diet contaminated with and without aflatoxins (*p*-value = 0.82). The symmetry and *p*-value indicate no evidence of publication bias.

**Figure 5 toxins-14-00707-f005:**
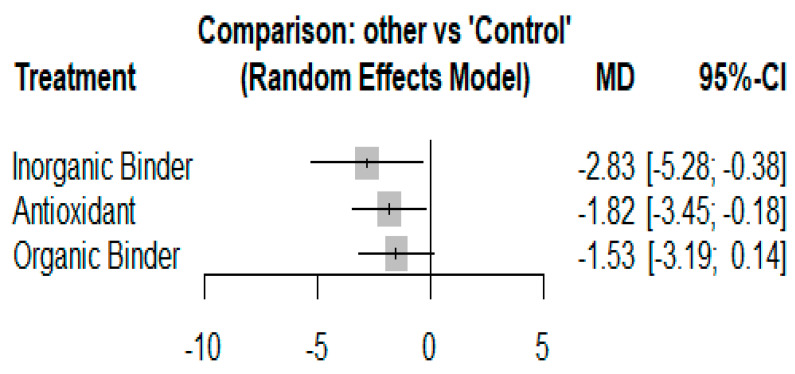
Forest plot of the efficacy of different additives (compared to control) to restore serum levels of alanine transferase in poultry birds exposed to aflatoxin-contaminated diets. Effects are presented as mean differences (MD), with negative values representing a reduction in the serum level of alanine transferase (higher negative value indicates more efficacy). The horizontal lines indicate 95% confidence intervals.

**Figure 6 toxins-14-00707-f006:**
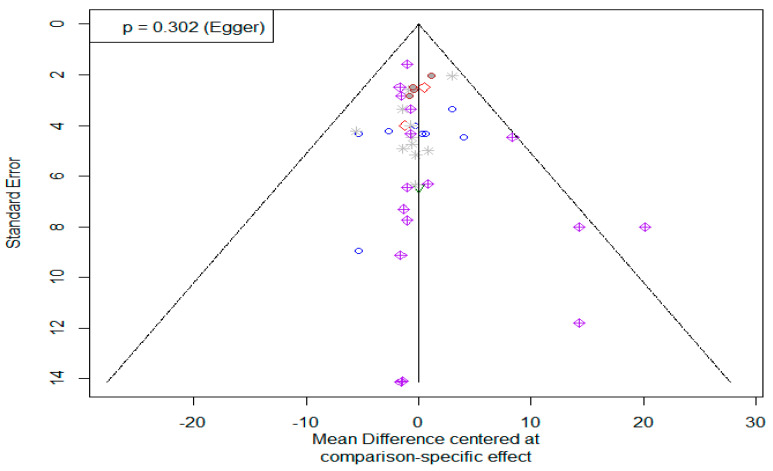
Funnel plot of network meta-analysis comparing the efficacy of different feed additives to mitigate the toxic effects of aflatoxin on poultry serum alanine transaminase. Egger’s test (*p* = 0.30) shows no evidence of publication bias.

**Figure 7 toxins-14-00707-f007:**
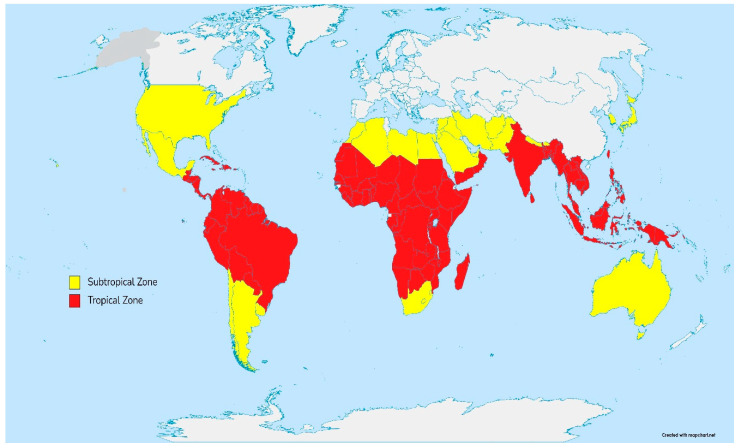
Map of the world indicating the tropical and subtropical zones.

**Table 1 toxins-14-00707-t001:** Background characteristics of studies included in network meta-analysis.

Reference	Country		Breed	Duration (Days)	Feed Additive		
					Type	Inclusion Level	Category
[19]	China		Broiler	21	Total flavonoids of Rhizoma Drynariae	125 mg/kg	Antioxidant
[20]	Iran		Broiler (Cobb 500)	35	L-Threonine	125% of the requirement	Antioxidant
[21]	China		Cobb broilers	28	*Penthorum chinense* Pursh extract	3 g/kg	Antioxidant
[22]	Iran		Ross broiler	42	Berberine	600 mg/kg	Antioxidant
[23]	Iran		Broiler (Ross 308)	42	Licorice extract	6 g/kg	Antioxidant
[24]	Iraq		Broiler (Ross 308)	35	Grape seed extract	200 mg/kg	Antioxidant
[25]	Mexico		Broiler	21	Curcumin	0.20%	Antioxidant
[26]	Pakistan		Broiler	42	*Chlorella pyrenoidosa* ethanolic extract	500 mg/kg	Antioxidant
[27]	Iraq		Broiler (Ross 308)	35	Levamisole	0.2 mL/kg	Antioxidant
[27]	Iraq		Broiler (Ross 308)	35	Vitamin E + Selenium	0.5 mL/L	Antioxidant
[28]	China		Cobb broiler	28	Grape seed proanthocyanidin extract	500 mg/kg	Antioxidant
[29]	China		Male arbor acre broiler	44	Sporoderm-broken spores of *Ganoderma lucidum*	200 mg/kg	Antioxidant
[30]	India		Male broiler	35	Carvacrol	1.00%	Antioxidant
[8]	China		Avian male broilers	35	Astaxanthin	10 mg/kg	Antioxidant
[31]	Iran		Broiler chicks (Ross 308)	42	Essential oil	500 mg/kg	Antioxidant
[32]	India		Broiler	42	Resveratrol	1.00%	Antioxidant
[33]	Pakistan		Broiler	35	Milk thistle	10 g/kg	Antioxidant
[34]	Thailand		Arbor acre broiler	42	Essential oil	0.50%	Antioxidant
[10]		Iran	Broiler chickens (Ross 308)	42	*Saccharomyces cerevisiae*	0.05%	Organic binder
[35]		Nigeria	Arbor acre broiler	30	*Saccharomyces cerevisiae*	2 g/kg	Organic binder
[36]		Mexico	Male broiler (Ross 308)	21	Yeast cell wall from *Saccharomyces cerevisiae*	0.05%	Organic binder
[23]		Iran	Broiler (Ross 308)	42	*Saccharomyces cerevisiae*	0.5 g/kg	Organic binder
[25]		Mexico	Broiler	21	Cellulosic polymer	0.30%	Organic binder
[37]		Egypt	Broiler	42	*Saccharomyces cerevisiae*	1 kg/ton	Organic binder
[7]		Iran	Broiler chicks (Cobb 500)	42	*Bacillus Subtilis* JQ618 strains	1 kg/ton	Organic binder
[7]		Iran	Broiler chicks (Cobb 500)	42	*Saccharomyces cerevisiae*’s cell wall	1 kg/ton	Organic binder
[8]		China	Avian male broilers	35	Esterified glucomannan	5 g/kg	Organic binder
[38]		Egypt	Broiler chicks (Ross 308)	28	Yeast cell wall	1%	Organic binder
[39]		Turkey	Broiler	21	Esterified glucomannan	1 g/kg	Organic binder
[40]		USA	Broiler	30	*Saccharomyces cerevisiae*	0.10%	Organic binder
[4]		China	Arbor Acres broilers	42	Smectite clay	2.5 kg/ton	Inorganic binder
[5]		Iran	Broiler chickens (Ross 308)	42	Aluminosilicate	2.5 g/kg	Inorganic binder
[41]		Egypt	Male Japanese quail chicks	42	Bentonite	1%	Inorganic binder
[41]		Egypt	Japanese quail chicks	42	Bentonite	1%	Inorganic binder
[23]		Iran	Broiler chickens (Ross 308)	42	Biochar	5 g/kg	Inorganic binder
[42]		Bulgaria	Toulouse geese	42	Aluminosilicate	0.5 g/kg	Inorganic binder
[24]		Iraq	Broiler (Ross 308)	35	Aluminosilicate	100 mg/kg	Inorganic binder
[7]		Iran	Broiler (Cobb 500)	42	Aluminosilicate	1 kg/ton	Inorganic binder
[7]		Iran	Broiler (Cobb 500)	42	Hydrated Sodium Calcium Aluminosilicate	15 kg/ton	Inorganic binder
[43]		Greece	Broiler (Ross 308)	42	Bentonite	1%	Inorganic binder
[43]		Greece	Broiler (Ross 308)	42	Bentonite	1%	Inorganic binder
[43]		Greece	Broiler (Ross 308)	42	Bentonite	1%	Inorganic binder
[33]		Pakistan	Broiler	35	Aluminosilicate	3 g/kg	Inorganic binder

## Data Availability

All data used in this study are available in the Figshare repository under the access number: 10.6084/m9.figshare.20145605.

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
