# Peer review of "The Efficacy of Additives for the Mitigation of Aflatoxins in Animal Feed: A Systematic Review and Network Meta-Analysis"

_toxins, 2022, doi:10.3390/toxins14100707_

Round 1
Reviewer 1 Report
Dear Authors,
The article submitted for review, entitled: “Comparative efficacy of additives for the mitigation of aflatoxins in animal feed: a systematic review and network meta-analysis” is very interesting. It summarizes the current knowledge on the detoxification capacity of various feed additives in poultry production.
In my opinion, the article should be approved for publication in the presented form.
My only suggestion is to supplement Table 1 with the aflatoxin dose in individual studies.
Regards
Author Response
The article submitted for review, entitled: “Comparative efficacy of additives for the mitigation of aflatoxins in animal feed: a systematic review and network meta-analysis” is very interesting. It summarizes the current knowledge on the detoxification capacity of various feed additives in poultry production.
In my opinion, the article should be approved for publication in the presented form.
My only suggestion is to supplement Table 1 with the aflatoxin dose in individual studies.
Regards
Thank you. The Inclusion of the aflatoxin concentration will make the tables very messy and too cumbersome. However, all the data extracted from cited references, including aflatoxin doses is freely accessible online for readers.
https://doi.org/10.6084/m9.figshare.20145605 and the link to this data is in the manuscript under data availability.
Reviewer 2 Report
The manuscript entitled "Comparative efficacy of additives for the mitigation of aflatoxins in animal feed: a systematic review and network meta-analysis" is a systematic review and network meta-analysis that compare and identify the most effective intervention to alleviate the negative impact of aflatoxins on the important livestock sector, poultry production. The authors report some information that aids to decide which type of additives to use for mitigating exposure to aflatoxins and thus improve food security and the livelihoods of smallholder farmers in developing countries. The information reported in the manuscript can be used in further studies. The manuscript seems technically ok. There are only some minor points to change.
1 - Figure 3.
The figure appears to be distorted. Please, improve the quality of figure 3.
2 - Figure 7.
Please clarify, Alaska is yellow-colored, which means that is a Subtropical Zone.
Please, clarify/correct.
Author Response
The manuscript entitled "Comparative efficacy of additives for the mitigation of aflatoxins in animal feed: a systematic review and network meta-analysis" is a systematic review and network meta-analysis that compare and identify the most effective intervention to alleviate the negative impact of aflatoxins on the important livestock sector, poultry production. The authors report some information that aids to decide which type of additives to use for mitigating exposure to aflatoxins and thus improve food security and the livelihoods of smallholder farmers in developing countries. The information reported in the manuscript can be used in further studies. The manuscript seems technically ok. There are only some minor points to change.
1 - Figure 3.
The figure appears to be distorted. Please, improve the quality of figure 3.
Thank you. The figure is not distorted, this is the standard forest plot produce by R staistical software
2 - Figure 7.
Please clarify, Alaska is yellow-colored, which means that is a Subtropical Zone.
Please, clarify/correct.
Thank you. The figure has been corrected.
Reviewer 3 Report
The article review entitled “Comparative efficacy of additives for the mitigation of aflatoxins in animal feed: a systematic review and network meta-analysis" presents a quite serious and rigorous study. The authors have done a great job. The study conducted by meta-analysis investigating the efficacy of different categories of feed additives in minimizing poultry exposure to aflatoxins is very interesting. This study makes it possible to compare the efficacy of three categories of feed additives. This study has revealed that the supplement based on inorganic binders provides the most protection among those tested. It is very important that the results of this study be made known in the regions most affected by aflatoxins. I have a few points to make. They are as follows:
1.- Check the page numbering starting on page 4.
2.- Figue 3.- In the first column, replace the author's name with the reference number. This makes it easier for the lector to read.
The article is publishable after these modifications have been made.
Author Response
The article review entitled “Comparative efficacy of additives for the mitigation of aflatoxins in animal feed: a systematic review and network meta-analysis" presents a quite serious and rigorous study. The authors have done a great job. The study conducted by meta-analysis investigating the efficacy of different categories of feed additives in minimizing poultry exposure to aflatoxins is very interesting. This study makes it possible to compare the efficacy of three categories of feed additives. This study has revealed that the supplement based on inorganic binders provides the most protection among those tested. It is very important that the results of this study be made known in the regions most affected by aflatoxins. I have a few points to make. They are as follows:
1.- Check the page numbering starting on page 4.
Thank you. This error is from toxins journal typesetting team, this will be corrected during production process/proofreading.
2.- Figue 3.- In the first column, replace the author's name with the reference number. This makes it easier for the lector to read.
Thank you. The respective references have been inserted in front of Authors’ name.
The article is publishable after these modifications have been made.